# Essential Components of Synthetic Infectious Prion Formation De Novo

**DOI:** 10.3390/biom12111694

**Published:** 2022-11-16

**Authors:** Kezia Jack, Graham S. Jackson, Jan Bieschke

**Affiliations:** MRC Prion Unit at UCL, Institute of Prion Diseases, Courtauld Building, 33, Cleveland Street, London W1W 7FF, UK

**Keywords:** prions, synthetic prions, PMCA, RT-QuIC, aggregation, infectivity

## Abstract

Prion diseases are a class of neurodegenerative diseases that are uniquely infectious. Whilst their general replication mechanism is well understood, the components required for the formation and propagation of highly infectious prions are poorly characterized. The protein-only hypothesis posits that the prion protein (PrP) is the only component of the prion; however, additional co-factors are required for its assembly into infectious prions. These can be provided by brain homogenate, but synthetic lipids and non-coding RNA have also been used in vitro. Here, we review a range of experimental approaches, which generate PrP amyloid assemblies de novo. These synthetic PrP assemblies share some, but not necessarily all, properties of genuine infectious prions. We will discuss the different experimental approaches, how a prion is defined, the non-protein requirements of a prion, and provide an overview of the current state of prion amplification and generation in vitro.

## 1. Introduction

Prion diseases, or transmissible spongiform encephalopathies (TSEs), are a class of infectious neurodegenerative diseases. Scrapie in sheep [1], CJD [2] and Kuru, amongst the tribes of Papua New Guinea practicing transumption [3], were some of the earliest TSEs identified in the 18th century, the 1920s and the 1950s, respectively. In 1959, a link was made between the diseases [4], and since then the class of prion diseases has expanded to include several other genetic diseases [5]. The infectious agent of scrapie, and so of the other TSEs, was isolated by fractionation from brain homogenate infected by scrapie-derived prions that were passaged into Syrian hamsters, and identified to contain no nucleic acids, but a single protein as its main component: PrP [6,7]. This protein is encoded in humans by the host *PRNP* gene, and the transcript undergoes post-translational modifications to form the mature PrP protein. The N-terminal signal sequence (residues 1–22) is removed by peptidases, and the C-terminal region (residues 231–254) is removed to allow the GPI anchor to be covalently attached. The protein can also be glycosylated at two sites (residues 180 and 196), allowing the production of mono-, di- or un- glycosylated PrP protein monomers [8]. In its native state the protein is located on the outer edge of the plasma membrane tethered by a glycosylphosphatidylinositol (GPI) anchor [9]. Its primary role is currently unclear, but it is highly conserved across all mammals [10].

Although TSEs are comparatively uncommon diseases, affecting one to two people per million each year worldwide [11], they have the potential to have profound societal impacts, as demonstrated by the BSE epidemic in the UK during the 1980s and 1990s and the more recent spread of chronic wasting disease in deer and elk in North America and elsewhere [12,13]. It is important, therefore, to understand the nature of the infectious agent and its replication to aid the prevention of future outbreaks and to develop therapeutic and prophylactic treatments and practical methods for inactivating the infectious agent. Furthermore, while authentic TSEs are rare, many other neurodegenerative diseases have been found to spread on the cellular level by prion-like mechanisms. This means that the misfolded forms of disease-associated proteins can act as templates for healthy protein to self-assemble into amyloid, which is postulated to be the mechanism of spread throughout the brain [6,14]. Highly ordered fibrillar amyloid structures are characterized by the 4.7–4.8 Å distance of intermolecular cross-β-sheets between stacked polypeptide chains and, historically, by the birefringence of the amylophilic dye Congo red [15]. The recently solved structures of two prions strains, 263K and RML, both have a parallel in register intermolecular β-sheet (PIRIBS) amyloid structures [7,16]. Prion strains are defined as having different disease characteristics (e.g., incubation periods) and PK digestion profiles [17], which are encoded in different amyloid folds. Most other disease-associated amyloid fibrils share the general PIRIBS architecture [18].

Prion-like mechanisms of templated amyloid formation are postulated to contribute to the pathobiology of other diseases including Alzheimer’s and Parkinson’s disease [19]. Whilst their templated self-assembly is believed to be similar if not identical to prion replication, they mostly lack acquired aetiologies in the course of the disease, although their pathology can be experimentally transmitted in animal models and in rare iatrogenic cases [20,21,22].

This review focuses on efforts to generate and amplify the infectious agent of TSE; in this context the term ‘prion’ therefore refers to assemblies of the prion protein PrP. There are two general approaches to investigating the makeup and formation of the infectious agent of TSEs: a top-down approach starting with the genuine disease agent and examining its behaviour and makeup in live systems or a bottom-up approach which attempts to mimic the disease agent starting with known components and conditions. While both approaches have merit, the advantage of the bottom-up approach is that much of the inherent complexity of living systems can be removed, which promises to isolate and identify the necessary components of the infectious agent. For this reason, there has been widespread effort to generate synthetic prions. It is not simply a technical problem to be solved, but rather will deepen our understanding of prion replication and so allow treatments to be developed. Figure 1 shows the overall methodology of the top-down and bottom-up approaches.

## 2. Defining a Prion

Prions are made up of alternatively folded conformers of the prion protein PrP, which form a cross-β amyloid as opposed to the α-helices found in the native PrP^C^ conformer. As discussed in the introduction, two prion strain structures recently solved have a parallel in register intermolecular β-sheet (PIRIBS) structure [7,16]. While the two strains show some subtle differences, they have the PIRIBS structure in common, so it is likely that having a PIRIBS structure is a requirement for a prion. Prions typically have some level of Proteinase K (PK) resistance, and are highly stable under various denaturing conditions [23]. Prions are defined by their biological activity: they must cause transmissible disease in animals. There are however many facets to the term ‘transmissible disease’, including infectivity and toxicity, which must be carefully considered. After all, to be able to determine if a synthetic prion has been made, one must first know what a prion is or, at the very least, what a prion is not. The three traits of infectivity, toxicity and PK resistance are a good place to start when discussing what is and is not a prion. Figure 2 shows how the definition of a prion involving these traits has changed since initial top-down prion studies as experiments in vitro have produced misfolded PrP species, which shared one or more characteristic traits of prions, whilst lacking infectivity in vivo.

### 2.1. PK Resistance

Native state PrP^C^ shows no PK resistance. Conversely, PK-treated preparations of infected brain homogenate contain high titres of infectivity, and PK-resistant, highly infectious assemblies of PrP (‘prion rods’) can be purified from these homogenates [17]. The core of the protein left after PK digestion of prions has been shown to vary slightly in molecular weight between different TSEs and disease strains [24], which were recently shown to correspond to distinct fibril structures [16,25]. Strains will be further discussed later in the article. However, inoculation of suitable experimental hosts with PK-digested and undigested scrapie brain homogenates have shown that a portion of infectivity is sensitive to PK digestion and reported to be in fractions as small as a PrP dimer [26]. Another challenge to this trait is that misfolded PrP conformers have been produced, which display PK-resistance, but are not classed as prions by any other measure [27]. This means that while some prions display PK resistance, not all PrP conformers that display PK resistance are prions.

### 2.2. Toxicity

As prions are infectious, it follows logically that they are also responsible for the toxicity seen in prion disease, and the neurological disfunction that eventually results in death. It has however been demonstrated experimentally that toxicity and infectivity can be separated; preparations of PK-digested prion rods that were shown to be highly infectious did not display any intrinsic toxicity in primary neurons [28]. To date, no synthetic PrP conformer has been produced that is definitely toxic in animals, although several claims to this have been made, which will be discussed later in Section 4. This means that not all prions are toxic, but that the ability to cause harm is essential for a prion to ultimately cause neurodegenerative disease. It could thus be argued that self-replicating misfolded PrP conformers that deposit in the brain but do not cause neurodegeneration lack an essential component of prion disease. It is important to make a distinction between prions being directly toxic and having a toxic effect, as current hypothesis suggest that it is not prions themselves that are toxic, instead another species is responsible for the toxicity of disease [28]. This species could be a by-product of prion formation, an off-pathway amyloid PrP structure, or a non-protein component whose formation is catalysed by prions [29].

### 2.3. Infectivity

The trait of infectivity is what sets TSEs apart from other neurodegenerative diseases that involve protein misfolding. This infectivity is not just observed experimentally, as can be demonstrated for other diseases such as Alzheimer’s [20,21,22], but is observed in the context of naturally occurring disease, such as in scrapie and chronic wasting disease. Infectivity is generally defined on the organismal level as the ability of a pathogen to invade and self-replicate inside a susceptible host. However, cell-based models, such as in the scrapie cell assay [30,31], are widely accepted proxies for measuring prion infectivity. It should be noted, though, that several amyloid species can replicate in cell models and be transmitted from cell to cell, despite not being infectious in the sense of the above definition [32,33]. True infectivity also requires continuity in the disease phenotype between the infected living systems—such as is observed in different prion strains. The protein-only hypothesis requires that all of the information required to establish a new and continuous infection in a new host must be contained within the protein [6]. Recent structural studies strongly support the view that it is encoded in the conformation of the PrP polypeptide chain in fibrillar prion assemblies [7,16]. True infectivity has not yet been reliably demonstrated for synthetic PrP conformers at titres comparable to authentic prions. However, inheritance of traits across sequential seeding rounds has been observed in synthetic systems [34], in what could be termed as in vitro infectivity.

## 3. Synthetic Prion Generation

When discussing what constitutes a synthetic prion, and how one can be defined, both the methods to generate synthetic prions and the methods to assess their biochemical and infectious properties must be established. These methods to generate and/or replicate prions in vitro must provide favourable conditions and environments for PrP to convert into the disease-associated state [35]. Figure 3 outlines the starting components, buffer conditions, physical environment and readout method of each of the four methods discussed here.

### 3.1. PMCA

PMCA (protein misfolding by cyclic amplification) was developed as a method to allow authentic prion replication in an in vitro system [36]. Similarly to the RT-QuIC, it starts with a prion sample, the ‘seed’, (purified or from tissue homogenate), but typically uses PrP^C^ from brain homogenate as substrate for amplification. Unlike the other assays discussed here, it does not provide real-time readout (as the presence of brain homogenate would interfere with the ThT fluorescence), instead the progression of the reaction is measured at the end point by Proteinase K resistant band intensity on a western blot. This method has been used successfully to amplify prions, which are infectious in vivo [36]. Serial rounds of PMCA amplification and dilution yield prions which statistically contain no PrP molecules of the initial seed and still retain the prion seed’s characteristics [37,38]. There is debate as to whether this is truly synthetic prion generation, if the starting material is a prion. PMCA may instead provide the necessary components and conditions for genuine prion replication, without being able to form prions de novo. The use of brain homogenate also means that there are many unknown components present in unknown quantities; this means it is not a bottom-up, but rather a top-down approach to prion replication, in which components required to form a prion need to be identified from a complex mixture. Some variations on the original PMCA approach address this shortcoming by using recombinant PrP instead of crude brain homogenate, which allows for components of brain homogenate to be added individually [39]. Alternatively, recombinant PrP can be used with PrP-null brain homogenate [40].

### 3.2. Amyloid Seeding Assay/RT-QuIC

In vitro assays for amyloid fibril formation initially attempted to recreate the de novo formation of prion fibrils [35,41,42,43]. The seeding capacity of amyloid fibrils has been exploited for the detection of prion infection in host animals and humans [44,45]. Amyloid seeding assays (ASA), including RT-QuIC, which are based on the self-replicating nature of prions [46,47,48,49] involve adding an initial amyloid seed to an excess of native monomer under conditions that are favourable to protein misfolding, such as elevated temperatures or kinetic perturbation. Amyloid fibril growth, by the addition of monomers to the initial seed, is tracked in real-time by using amyloidophilic dyes such as thioflavin T (ThT), which show an increased fluorescence when bound to fibrils. The method was adapted to detect low volumes of prions in a sample, and these seeded aggregation assays were further developed into the RT-QuIC assay (real-time quaking induced conversion) [49,50], which is widely used for diagnostics today. These assays amplify small amounts of starting prions into a detectable readout through the use of fluorescent tags or amylophilic dyes [51]. RT-QuiC excels at sensitivity and can produce amyloid from minute amounts of seed down to a single seed particle [51]. However, since RT-QuIC assays were developed as diagnostic tools, and while they may help to distinguish prion strains [52] they do not aim to faithfully replicate prion structure, but rather to amplify PrP amyloid from a range of starting prion samples [53]. Most of the work in developing the RT-QuIC aims to improve the sensitivity of the assay and reproducibility of results [54]. Future structural studies will determine to which degree conformers formed in the RT-QuIC are identical or distinct from authentic prions.

### 3.3. Semi-Denaturing Amyloid Seeding Assays

The semi-denaturing seeding assay uses similar physical conditions as the RT-QuIC to accelerate fibril growth, i.e., shaking of recombinant PrP substrate with prion seed in a buffer with zirconium beads. It does not involve the elevated temperatures of RT-QuIC, but instead uses chaotropes in the buffer to partially denature the PrP^C^ substrate. The method uses the full length PrP or a fragment of the full protein sequence [55]. This method, or variations on it, has been widely used in the literature to produce a range of PrP conformers [44]. Some of these conformers have been described to be infectious in animal models [42], albeit with very low specific infectivity. The assay provides a real-time fluorescence readout to track fibril growth, so the kinetics of different conditions can be analysed and compared. There is some debate as to how the initial denaturation of the monomer and the denaturing conditions affect the final structure and so properties of the fibril, and the assembly mechanism. One theory is that in the physiological disease process prions are formed in the low pH environment of the lysosome, which is mimicked by the presence of chaotropes and may partially denature PrP [56], but this is debated [57].

### 3.4. Native Aggregation Assays

The limitations of the above assay with regard to the denaturing conditions, and the concerns that the method does not well represent the physiological disease process, led to the development of PrP aggregation assays under near-native conditions [58]. In essence the only difference between this and the above method is the absence of any denaturant, so the protein monomer is kept in its native state. Early approaches required the presence of detergents at low concentrations [35]. Native aggregation assays in the absence of detergents require careful optimisation of reaction conditions to prevent protein precipitation whilst maintaining growth rates to fit feasible experimental time frames. We recently demonstrated that the native aggregation assay can be used to grow PrP amyloid under conditions that are near-physiological [58]. It is thought that this assay will most closely model the mechanism of growth as well as protein–protein interactions that take place in genuine disease.

## 4. Non-Protein Requirements

It is proven that proteins rather than nucleic acids encode the information replicated during prion infection, as infectivity is resistant to heat, formalin, solvents, nucleases, UV and ionising radiation [6]. This however does not mean that other components are not involved in the formation or growth of prions, or indeed are present as structural components within a prion [59]. Several classes of compounds have been identified that seem to support the growth or amplification of genuine prions experimentally. The fact that only PMCA from crude brain homogenate has successfully amplified prions to biologically comparable infectivity titres provides further evidence for this. The field of co-factors in prion replication is a large one, so this section will only briefly cover the main points. With synthetic prions, strains differ from each other mainly by the weight of their PK resistant core, and also by the disease symptoms they cause, and the incubation time [60]. It has been suggested that the co-factors present influence strain selection and characteristics, and the correct co-factors are required for propagation of different strains [60].

### 4.1. Post-Translational Modifications

As previously stated, the substrate for a PMCA reaction originally was crude brain homogenate, but now can also consist of purified endogenous PrP^C^ or recombinant PrP^C^. PrP^C^ can be purified from crude brain homogenate by detergent solubilization, Protein A agarose, PrP immunoaffinity, and cation exchange chromatography [39]. The PrP^C^ from this method is co-purified with equimolar quantities of 20-carbon fatty acids, which are not bound covalently to the protein. Saponification of crude brain homogenate has been shown to eliminate prion propagation [61], so it follows that lipids play a crucial role in prion propagation. Bacterially derived recombinant PrP (rPrP), while having the same amino acid sequence as endogenous PrP, does not contain any post-translational modifications. Namely, there is no asparagine N-linked glycosylation and no GPI anchor on rPrP molecules [62]. Whilst prion strains isolated ex vivo each display characteristic ratios of un- mono- and di-glycosylated PrP, the recent structural data do not yet offer a compelling explanation for these glycosylation patterns [7,16,25]. There have been reports of infectious prion generation from rPrP [63], implying that these post translational modifications are not required for prion formation. However, the efficiency of this conversion is very low, such that some labs report no infectious prion generation from the same substrate [64].

### 4.2. Lipids and RNA

In vitro prion replication is more efficiently supported by brain homogenate than by purified or recombinant PrP^C^, suggesting some role for co-factors in the propagation of prions, perhaps as an additional component or to stabilise certain conformations [65]. It has been shown that RNA and lipids alone can support the propagation of synthetic prions [66], which naturally raised questions as to whether the nucleic acid played a role in encoding genetic information. Using synthetic polyriboadenylic acid in PMCA reactions has confirmed that no genetic informational RNA is required to support the propagation of prions in vitro [59,67], which excludes the possibility that TSEs are caused by a virus. The co-factor activity is likely due to structural properties of the RNA interacting with the PrP. The latest high-resolution cryo-EM structures of RML prions have discovered a positively charged patch on the exposed surface of the prion [7] to which negatively charged RNA or other polyanions could bind. This binding could play a structural role in stabilising the resultant structure.

More recently, it has been demonstrated that there is a nuclease-resistant co-factor activity present in brain homogenate. Using purification and reconstitution experiments it was shown that this molecule is phosphatidyl-ethylanolamine (PE), which can act as a sole co-factor whether supplied endogenously or synthetically derived [61]. The synthetic phospholipid POPG (1-palmitoyl-2-oleoyl-sn-glycero-3-phospho-(1′-rac-glycerol)) has also been shown to support prion replication in reconstitution experiments, but only in combination with the poly-anion RNA [66].

Co-factors have been shown to play a role not only in effective propagation, but also in maintaining strain differences. By performing PMCA reactions with different combinations of RNA and POPG or with PE as the single co-factor, it was found that the same strains would form from the same co-factors. Additionally, different seeds propagated with the same co-factor would converge into the same strain. This goes some way to explaining how a single polypeptide sequence can exist stably in many different conformations; the presence or absence of different co-factors serve to influence the tertiary structure of the protein [68].

## 5. Current State of the Field

Table 1 shows a selection of the efforts to produce synthetic prions de novo. It is not exhaustive, but aims to cover the main methods and co-factors used. Some of the reactions produced PK-resistant PrP conformers that did not cause disease in vivo [64,66]. Interestingly, some animals without any clinical symptoms or detectable PrP deposition showed positive results in an RT-QuIC assay. This further supports the hypothesis discussed in Section 2.2 that prion replication and prion-disease toxicity may involve distinct molecular species and pathways, as the positive RT-QuIC demonstrates that there was seeding and aggregate growth in vivo without any toxic effects.

It has also been shown that reactions under the same conditions with the same components can cause different incubation periods in animals [76]. This suggests that different seeding competent species, i.e., different strains, were produced. Subtle difference in sonication power output could possibly cause these differences, or they could be a result of the inherently stochastic nature of the aggregation process favouring the replication of the structure that was established first [75]. It has been observed that less efficient sonication horns facilitate prion formation in purified substrates better than crude brain homogenate, while newer high-energy horns facilitate formation better with crude brain homogenate [70]. Recent structural studies on other amyloidogenic proteins have discovered a large number of structural polymorphs [77], which has reinforced the hypothesis that prions may exist as a quasispecies [78], i.e., a population of co-existing conformers generated by imperfect replication of the structural information of the prion [79,80]. It is plausible that, analogous to evolutionary fitness on the genetic level, different conformers could be selected for in replication under subtly different conditions.

Other work has reported the formation of recombinant prions from truncated (91–231) recombinant PrP, which cause disease only in transgenic mice overexpressing truncated PrP [81]. The use of truncated substrate also suggests that the synthetic prions produced may not be formed in the same way as during the disease process, so the findings may not be transferable to the wild-type disease. A large variety of different animal lines and genetic backgrounds are used to score infectivity and serial passage; some of these lines overexpress PrP, or express a transgenic version, so do not well represent true disease. Some lines also form prions spontaneously, as noted in a number of papers where the control animals display protease-resistant PrP [68]. Recent advances in analysing prion and amyloid structures by cryo electron microscopy promise to shed light on the structures of PrP fibrils generated by different protocols and their overlap with the structures of prion rods isolated ex vivo. Whilst there is no certainty that any co-factors required for prion propagation may be incorporated into a final prion assembly, it is intriguing to note there were poorly defined electron densities in the structures of both the 263K and RML prions recently determined [7,16].

### Limitations

The table shows that synthetic prions have indeed been successfully produced de novo. They are de novo as they start without any initial prion seed, and are synthetic because they are made from recombinant protein monomer and artificial, not brain, derived co-factors. None of this work has been conducted with human systems to date (although human brain homogenate was tested in one paper and found to produce no protease resistance de novo [71]). It would be valuable to test the most promising co-factors and mechanisms with human rPrP or brain homogenate, although this work is difficult to carry out due to safety considerations. The use of the synthetic lipid POPG and liver RNA leaves open the question of which natural components fill these roles in the authentic disease process.

Another limitation of much of the above work is the low specific infectivity of synthetic prions. Purification of bona fide prions from infected brain homogenate yields specific infectivity of ~10^9^LD_50_ units/mg PrP [17]. The specific infectivity of the majority of synthetic prions is variable and can be up to a million-fold lower than obtained from brain homogenate. This could mean that the de novo material in fact templates prion conversion poorly, so is structurally distinct from bona fide prions. Another explanation would be that only a small proportion of the final material is able to cause disease. To distinguish between these two, or indeed other, possibilities needs assessing how homogenous or heterogenous the material formed is, which requires the structural characterization of single prion fibers in a population.

While it is an achievement to have defined a set of components that produce synthetic prions, the fact that the same components under the same conditions do not always produce synthetic prions shows that the bottom-up approach, whilst sufficient to support the protein-only hypothesis, has not yet been perfected. In the genuine bottom-up approach, this source of variation must be understood, as otherwise there are still unknown factors in prion formation. It is likely that the physical conditions (e.g., sonication) are responsible for some of this variation. A large-scale study has systematically investigated a wide range of environmental conditions [69], and while this work could not reproduce the generation of infectivity or toxicity—likely due to the experimental design for fibril formation lacking effective fibril fragmentation by agitation, sonication or beads—the approach is a valuable one that should be repeated. More work will be needed into the reproducibility of the results and to identify the specific mechanistic factors that influence prion formation

From here, the next step in the field of synthetic prions may be to form bona fide prions de novo, containing the same components as a genuine prion. This would provide a link between the top-down and bottom-up approaches of investigating prions, and allow therapeutic and prophylactic treatments to be developed.

## Figures and Tables

**Figure 1 biomolecules-12-01694-f001:**
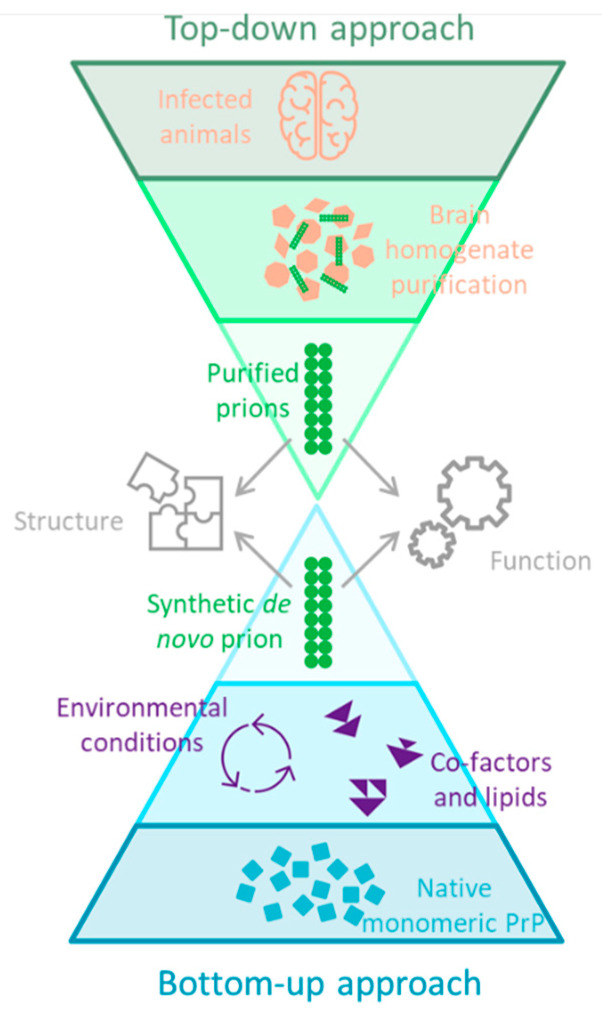
An outline of the top-down and bottom-up approach for investigating prions. The top-down approach has largely been achieved, with the high resolution structures of two prion strains having been recently solved [17], as well as many more structures of other amyloid classes [18]. The bottom-up approach starts from simplified components and aims to produce a prion with the same structure and biological activity as the brain-derived prions. This approach requires identification of any potential co-factors, and so provides further understanding of the disease process. It allows for modulation of the process by adding potential inhibitors or accelerators, and so contributes to the development of therapeutics.

**Figure 2 biomolecules-12-01694-f002:**
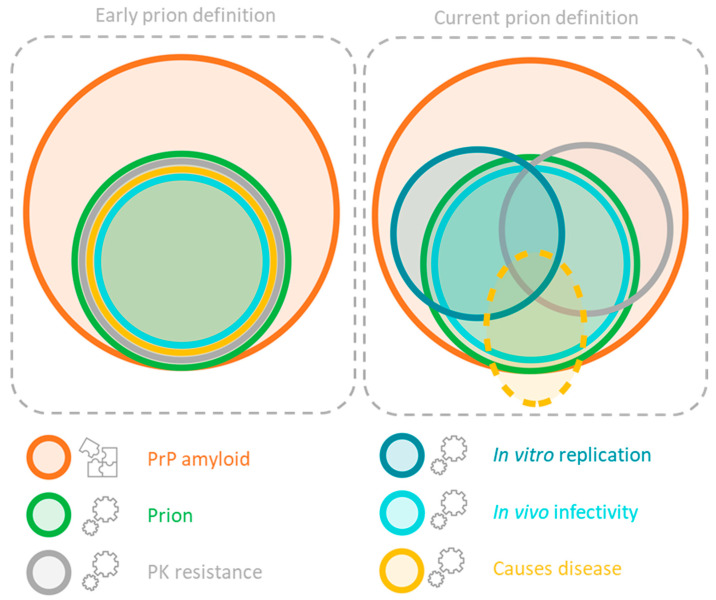
Whilst the core of the definition of a prion as an infectious protein structure as stated by Prusiner has remained, the boundaries of its definition have since blurred as protein assemblies, which are not infectious in vivo, can possess traits that were thought to be specific for prions. PrP amyloid (orange) is defined by is structure, a PrP protein sequence with a cross-β-sheet structure, whereas a prion (green) is defined by its biological activity (light blue). All solved structures of bona fide prions are PIRIBS amyloids, suggesting that amyloid can contain prion-like traits. However, biochemical properties such as PK resistance (grey) and in vitro replication (dark blue) that are used as proxies for the detection of prions are shared by amyloid assemblies, which are not infectious in vivo. The dashed line (yellow) signifies that much is still unknown about how prions cause disease—the toxicity may be caused by a non-protein species downstream of the prion, as well as by the prion itself.

**Figure 3 biomolecules-12-01694-f003:**
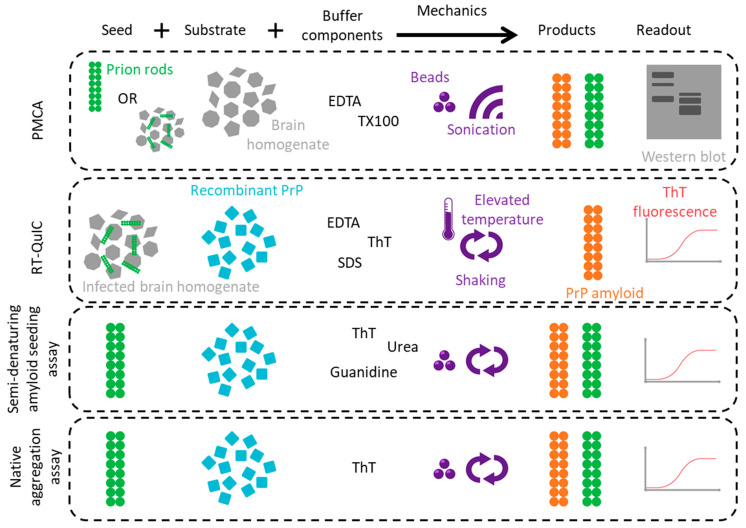
In vitro prion amplification assays. A direct comparison of the main in vitro aggregation methods, with regard to their respective components, conditions and outputs. Variations with regard to the starting seed and substrate, as well as how the final product is defined, are further discussed in the relevant section, but shown here is the typical set up for each.

**Table 1 biomolecules-12-01694-t001:** Comparison of the main reports of de novo synthetic prion generation over the last twenty years. Further details, including buffer conditions, specific infectivity and incubation period (where available in the papers) can be found in the supporting information. * This high specific infectivity has to date not been replicated using the same conditions [69].

Year	Name of Strain	Seed	Substrate Source	Co-Factors	Aggregation Mechanism	Seeding Ability	Animal Strain	Specific Infectivity
**2000** **[70]**		None	Artificially synthesized PrP (89–143, Pro101Leu) peptide	None	3-week acetonitrile incubation at 4°, followed by 3 lyophilisation and washing steps	N/A	Mice expressing low levels of a transgene coding for PrP (Pro101Leu)	Not mentioned
**2004** **[42]**	MoSP1	None	Recombinant, bacterial, Mouse PrP 89–230	None	Denaturing aggregation assay	Serial transmission to FVB mice and Tg4053 mice gave mean incubation time of 154 and 90 days, respectively	Mice overexpressing PrP (89–231) (known as Tg9949 mice)	Not mentioned
**2007** **[39]**		Sc237 PrP 27–30	Purified hamster PrP	Co-purified lipid, poly(A) RNA	PMCA	N/A	Golden Syrian hamsters	~5 × 10^4^ LD(50) per mL
		139H	Purified hamster PrP	Co-purified lipid, poly(A) RNA	PMCA	N/A	Golden Syrian hamsters	Not mentioned
		None	Purified hamster PrP	Co-purified lipid, poly(A) RNA	PMCA	N/A	Golden Syrian hamsters	~5 × 10^3^ LD(50) per mL
**2009** **[71]**		None	Sonicated brain homogenate from Syrian hamsters	Sonicated brain homogenate from Syrian hamsters	PMCA	Serial passage being carried out at time of publication	Syrian hamsters	Not mentioned
**2009** **[72]**	“MoSP5”, “MoSP6”, and “MoSP7”	None	Recombinant, bacterial, Mouse PrP 89–230 (MoSP6 and MoSP6) or Mouse PrP 23–230 (MoSP5)	None	Denaturing aggregation assay	Brain homogenates containing MoSP5, MoSP6, and MoSP7 transmitted disease to healthy Tg4053 mice ([72], Figure 3A and Table S5); MoSP6 and MoSP7 also transmitted disease to wild-type FVB mice	Mice overexpressing full-length, wild-type PrP (known as Tg4053 mice)	Not mentioned
**2010** **[63]**	rPrP-res(RNA)/OSU strain	None	Recombinant, bacterial, Mouse PrP 23–230	RNA (mouse liver), POPG	PMCA	Able to propagate with normal mouse brain homogenate PMCA	CD-1 mice	Not mentioned
**2010** **[73]**	SSLOW	None	Recombinant, bacterial, Golden Syrian Hamster PrP 23–231	Fibrils annealed with normal brain homogenate (with sonication)	Denaturing aggregation assay	Carried out serial passage, with some controls also producing PK resistant material	Golden Syrian hamsters	Not mentioned
**2010** **[74]**		MoSP1	Recombinant, bacterial, Mouse PrP 89–230	Co-purified from PTA prion precipitation (seeds)	Denaturing aggregation assay	Serial passage carried out caused disease	Tg9949 mice	Not mentioned
**2012** **[61]**		rPrP-res(RNA)	Recombinant, bacterial, Mouse PrP 23–231	Synthetic PE	PMCA	Propagation in many rounds of sPMCA	C57BL/6 mice	Not mentioned
**2012** **[67]**		rPrP-res(RNA)	Recombinant, bacterial, Mouse PrP 23–230	poly(rA) RNA, POPG	PMCA	Able to infect neuronal CAD5 cells	CD-1 mice	Not mentioned
**2012** **[68]**	OSU co-factor PrP	rPrP-res(RNA)	Recombinant, bacterial, Mouse PrP 23–230	Purified PE (mouse brain)	PMCA	Propagation of an ~18kD conformer maintained indefinitely	C57BL/6 mice	∼2.2 × 10^6^ LD50 units/μg PrP *
	OSU protein-only PrP	OSU co-factor PrP	Recombinant, bacterial, Mouse PrP 23–230	None	PMCA	40%: no propagation, 60%: adaption to ~16kD band (which can be propagated indefinitely with rPrP), no propagation with normal BH	C57BL/6 mice	N/A
**2013** **[64]**	rPrP-res(NIH)	None	Recombinant, bacterial, Mouse PrP 23–230	RNA (mouse liver), POPG	PMCA	No prion formation in scrapie susceptible cell lines (SN56 or CF10)	C57BL/10 mice	N/A
**2013** **[75]**	Same as 2010, but in prion-free lab	None	Recombinant, bacterial, Mouse PrP 23–230	RNA (mouse liver), POPG	PMCA	Able to propagate with normal mouse brain homogenate PMCA	CD-1 mice	Not mentioned
**2015** **[76]**		None	Recombinant, bacterial, Mouse PrP 23–231	None	Denaturing aggregation assay	Able to seed mouse hypothalamic GT1 cells and mouse neuroblastoma N2a cells	CD-1 mice	N/A
**2017** **[66]**	rPrP-res(RNA-low)	rPrP-res (RNA)	Recombinant, bacterial, Mouse PrP 23–230	RNA (mouse liver), POPG	PMCA	No prion formation in CAD5 cells, but able to seed RT-QuIC reaction	C57BL/10 mice or Tga20 mice (which overexpress PrP 23–231)	N/A
**2019** **[60]**		OSU co = factor PrP	Bank vole brain homogenate	Bank vole BH	PMCA	PMCA product propagates at 27–30 kD	M109 bank voles	Not mentioned
		OSU protein-only PrP	Bank vole brain homogenate	Bank vole BH	PMCA	PMCA product propagates at 27–30 kD	M109 bank voles	N/A

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
