# Peer review of "Essential Components of Synthetic Infectious Prion Formation De Novo"

_biomolecules, 2022, doi:10.3390/biom12111694_

Round 1

Reviewer 1 Report

The authors described in this review that how closely the prion-like protein in a bottom-up approach resembles the prion producted in Animals and Human. The authors described the contains very politely and concisely. This review will give most worth information future prion research. Unfortunately, the table is hard to see. Please, adjust the width of the columns to make the table easier to read.

Author Response

Reviewer 1

We would thank this reviewer for their supportive comments. Their only suggestion is to improve the clarity and presentation of Table 1. This has been modified as requested.

Reviewer 2 Report

In this superficial review Jack et al list efforts to form prions from soluble PrP. The review does not provide a lot of insight. Although most of the review deals with amyloid formation the uniqueness of amyloid is never clearly explained.

Minor comments

Line 23                  what is so unique about prion infections?

Line 28                  Prusiner used brains from scrapie infected hamsters.

Line 29                  Purified virus preparations likely have coat proteins as the main component.

Line 31                  Should it be known where GPI stands for?

Line 41                  From what is presented it is unclear what “prion-like mechanisms” mean.

Line 42                  These disease associated proteins are not just misfolded, they are folded into

amyloid.

Line 52                  From the remaining part of the review it is evident that no clear conclusions can easily

be drawn.

Figure 1                                The choice to represent amyloid by stacks of circles is very unfortunate. Amyloid is

formed by planar stacking of protein monomers.

Line 61                  Amyloid needs to be properly introduced.

Line 81                  RT-QuIC was developed after PMCA.

Line 88                  RT-QuIC assays have been shown to faithfully preproduce structural differences of prion

strains.

Line91-92             The authors clearly have a grudge against RT-QuIC. The statement made here is an

exaggeration.

Line 97                  Only RT-QuIC has been discussed up to this point so what is the other method?

Line 100                Prion strains are not only assessed based on proteinase K resistance patterns! Up to this

point it has not been explained what prion strains are. Prion strains were first described

based on clinical observations.

Line 116                As amyloid is not explained the reader will not be able to understand what “fibril

growth” means.

Line 119                OR variations

Line 142                A prion is not defined as an alternatively folded conformer of PrP! A prion is an

infectious protein.

Line 146                Same comment as line 142. Prions are not defined by their ability to cause disease in

animals.

Line 150                The word prion has been used a lot up to this point. It seems that what is and what is

not a prion should have been explained much earlier.

Figure 3                                This figure needs a better explanation.

Line 154                What was the original definition of a prion? What is the current definition of a prion?

Line 157                To my knowledge there is only one protein that has a fuction in the prion form, the

[Het-s] prion of Podospora anserina. To cause disease is (with some exceptions) not a function! The recently proposed prions [SMAUG+] and [BIG+] of yeast might also be functional prions.

Line 177                Being infectious is the definition of a prion.

Line 183                I did not find an in-depth about this claim in the paper.

Line 188                What is this evidence?

Line 197                Using a word to define a word.

Line 203                I think that most will agree that the protein nature of the scrapie agent is not an

hypothesis anymore.

Line 213                Infectivity being based on protein is the definition of a prion. Using the words “widely

accepted” indicates that the authors still have doubt?

Line 238                Both anchorless PrP as well as PrP lacking glycans can propagate scrapie in mice. The

prescence of these modifications should be explained.

Line 246                Nucleic acid having a genetically coding function in the TSEs was ruled out way before

the listed experiments.

Line 254                This sentence is chopped off.

Lines 255-261     Reference 70 should be part of this paragraph.

Line 269                Now that it has been firmly established that PrP forms parallel in-register beta amyloid

the structural properties of the amyloid fold should be mentioned. Although the cryo-

EM structures published to date show only relative minor variations the amyloid fold from some strains could still differ significantly.

Table 1                  Table 1 is not correctly formatted.

Line 291                It is not explained what structure is being replicated.

Line 298                Same issue as line 291.

Line 308                The test normally is to see if WT animals can be infected with material obtained from

the first or second passage in mutant animals.

Line 340                The data from reference 70 are not correctly described. Infectious material was

generated but any one condition identified could not reproducible do so.

Author Response

Reviewer 2

The reviewer made several useful suggestions, which we adapted to improve the manuscript. However, we found many of the reviewer’s comments obtuse and confused, and it seems that the reviewer has misunderstood both the scope of the review and the accepted view on prion biology, from which our review does not differ in any meaningful way. As has recently been demonstrated by our colleagues in the MRC Prion Unit and others, purified prions have the cross-beta sheet structure and the tinctorial properties that that are the defining properties of amyloid and indeed share the PRIRBS architecture of many other amyloid species. It is, however, equally clear that prions, unlike other amyloid species, uniquely cause transmissible disease in humans and animals. While it is easy to generate fibrils with amyloid properties from PrP, the generation of bona fide prions in vitro with high specific infectivity has eluded the field for decades. The aim of our manuscript is to review these efforts past and present and highlight their progress and challenges. In no way do we have it ‘out for’ any assay as the reviewer suggests, we merely review which assays capture which specific aspects of amyloid and/or prion formation.

Please find our point-by point response to the reviewer’s specific comments below:

Line 23 what is so unique about prion infections?

We have modified this sentence for clarity. They are unique in that they can infect a mammalian host, whereas, although amyloids may replicate, they do not constitute infections resulting in disease.

Line 28 Prusiner used brains from scrapie infected hamsters.

Yes he did. They were homogenised and fractionated to isolate the scrapie agent. This has been clarified in the manuscript.

Line 29 Purified virus preparations likely have coat proteins as the main component.

Yes they do. However, they also have nucleic acid components that code the genetic information of the virus, which the isolated protein (‘scrapie agent’) did not.

Line 31 Should it be known where GPI stands for?

This has now been defined.

Line 41 From what is presented it is unclear what “prion-like mechanisms” mean.

This is discussed in the citation we provided. However, we have expanded this sentence for clarity.

Line 42 These disease associated proteins are not just misfolded, they are folded into amyloid.

As above, prions are folded into PIRIBS structures which also possess the tinctorial properties of amyloid. However, there are an ensemble of misfolded structures associated with prion infection and replication, including non-prion and non-amyloid forms such as the putative PrPL structure.

Line 52 From the remaining part of the review it is evident that no clear conclusions can easily be drawn.

This is rather confusing. Conclusions can be drawn by defining the essential cofactors and components of prions and prion replication. However, this has not yet been achieved.

Figure 1 The choice to represent amyloid by stacks of circles is very unfortunate. Amyloid is formed by planar stacking of protein monomers.

This is a schematic and the important visual point is to distinguish starting monomers from ordered assemblies. We would argue that no geometric shape adequately represents the actual PrP fold in the prion fibril.

Line 61 Amyloid needs to be properly introduced.

This was a valid and useful critique. We have modified the test to make it clear amyloid is a self-replicating structure. We have also rearranged the structure of the review, so that amyloid and prions are now defined before discussing the different assays for amyloid and prion formation.

Line 81 RT-QuIC was developed after PMCA.

Yes it was, while other assays for the formation of PrP amyloid in vitro preceded PMCA. We are not clear what the reviewer is suggesting here.

Line 88 RT-QuIC assays have been shown to faithfully preproduce (sic) structural differences of prion strains.

This statement is incorrect as QuIC reaction products are not infectious. RTQuiC and other assays can however propagate amyloid fibrils with distinct structures. We have clarified this in the text.

Line91-92 The authors clearly have a grudge against RT-QuIC. The statement made here is an exaggeration.

We are not clear what the referee is asking or stating here? What is an exaggeration? This is a clear statement of fact and there is no suggestion of a grudge against RT-QuIC, which is an assay that has a very specific use at which it excels.

Line 97 Only RT-QuIC has been discussed up to this point so what is the other method?

ASA – Amyloid Seeding Assay developed by David Colby in Stanley Prusiner’s laboratory and is discussed and cited earlier in the manuscript.

Line 100 Prion strains are not only assessed based on proteinase K resistance patterns! Up to this point it has not been explained what prion strains are. Prion strains were first described based on clinical observations.

This is obviously true, since the differences in clinical phenotype and incubation times in bioassays are integral to the definition of a prion strain.  ! However, we have amended this sentence as the role of Western blotting will be obvious to most readers and is by far the most widely used assay to distinguish prion strains biochemically.

Line 116 As amyloid is not explained the reader will not be able to understand what “fibril growth” means. 2

As commented above, we have restructured the manuscript, so that amyloid and prion definitions precede any discussion of assays. We have also amended section 1.1 (now 2.1) and this now describes the principles of amyloid replication and detection using benzothiazoles.

Line 142 A prion is not defined as an alternatively folded conformer of PrP! A prion is an infectious protein.

This is again a confusing comment from the reviewer. The sentence reads ‘Prions are made up of alternatively folded conformers of the prion protein PrP’. This is not a definition and as a statement is correct.

Line 146 Same comment as line 142. Prions are not defined by their ability to cause disease in animals.

Prions are ‘proteinaceous infectious agents’ as defined by Prusiner and that is exactly what the manuscript states. The sentence in the manuscript reads ‘Prions are defined by their biological activity: they must cause transmissible disease in animals.’ Both the infectivity in vivo and the proteinaceous nature of the infectious agent are integral to the definition of a prion.

As above, we are not clear what the reviewer is trying to say.

Line 150 The word prion has been used a lot up to this point. It seems that what is and what is not a prion should have been explained much earlier.

As commented above, we have restructured the manuscript, so that the definition of a prion precedes the discussion of assays for prion replication as per the reviewer’s suggestion.

Figure 3 This figure needs a better explanation.

We agree that the figure needed more explanation to avoid being confusing. The figure illustrates that different properties (in vitro replication, PK resistance, detergent insolubility, amyloid fold) which were initially believed to be exclusive to prions have since been demonstrated in other protein assemblies that are not infectious in vivo. We have expanded the figure legend and the discussion in the text to make this point more clearly.

Line 154 What was the original definition of a prion? What is the current definition of a prion?

Please refer to our response to the comment above and to the comment on l. 146

Line 157 To my knowledge there is only one protein that has a fuction in the prion form, the [Het-s] prion of Podospora anserina. To cause disease is (with some exceptions) not a function! The recently proposed prions [SMAUG+] and [BIG+] of yeast might also be functional prions.

We are not clear what the reviewer is referring to here. While both functional amyloid and functional prions exist, that is not integral to the definition of a prion.

Line 177 Being infectious is the definition of a prion.

We have amended this sentence for clarity.

Line 183 I did not find an in-depth about this claim in the paper.

We have amended this sentence to signpost the relevant sentence clearly.

Line 188 What is this evidence?

Citations have now been included.

Line 197 Using a word to define a word.

We are not clear what the reviewer is referring to here.

Line 203 I think that most will agree that the protein nature of the scrapie agent is not an hypothesis anymore.

We obviously agree with the reviewer that the protein-only hypothesis is strongly supported by experimental evidence. The word ‘hypothesis’ does not imply a lack of experimental support, and the historic alternative virus hypothesis has been comprehensively disproved. However, it is still not clear what is responsible for the phenotypic differences in strains. This may or may not be encoded by the differences in prion conformation alone. Alternatively, it may be attributable to as yet unidentified co-factors.

Line 213 Infectivity being based on protein is the definition of a prion. Using the words “widely accepted” indicates that the authors still have doubt?

As we have stated above; the origin of strains is not known and may or may not be due to the protein conformation of prions alone. We certainly share the widely accepted view on prions as proteinaceous infectious particles.

Line 238 Both anchorless PrP as well as PrP lacking glycans can propagate scrapie in mice. The prescence of these modifications should be explained.

We are unclear what the referee is asking for here. Yes, they can propagate with low efficiency implying they are not required for prion formation. Which is precisely what is discussed in our manuscript.

Line 246 Nucleic acid having a genetically coding function in the TSEs was ruled out way before the listed experiments.

This is correct. We have amended the sentence for clarity.

Line 254 This sentence is chopped off.

This typographical error has been corrected.

Lines 255-261 Reference 70 should be part of this paragraph. 3

We have included this citation here.

Line 269 Now that it has been firmly established that PrP forms parallel in-register beta amyloid the structural properties of the amyloid fold should be mentioned. Although the cryo-EM structures published to date show only relative minor variations the amyloid fold from some strains could still differ significantly.

We have added a brief description of the recent prion structures to section 2.0 and described the evident differences in atomic structure between strains. The structure of prions, however, is not the subject of this review.

Table 1 Table 1 is not correctly formatted.

We have reformatted Table 1 to make this clearer to read.

Line 291 It is not explained what structure is being replicated.

The replicating structures are not described by the authors of the original study. We have amended this sentence to describe them as ‘species’ rather than structures.

Line 298 Same issue as line 291.

These are amyloid polymorphs as stated in the sentence in the manuscript and described in the cited paper.

Line 308 The test normally is to see if WT animals can be infected with material obtained from the first or second passage in mutant animals.

We assume that the reviewer is referring to the standard test for infectivity in vivo.  We fully agree with the reviewer on this point. It does not, however, address concerns that the fibril assembly mechanism in ASA may be different from that of authentic prions in WT animals and that the same may be true in animal models that overexpress truncated PrP.

Line 340 The data from reference 70 are not correctly described. Infectious material was generated but any one condition identified could not reproducible do so.

This is not correct. The results did not indicate the definite production of infectivity in any condition. We would point out that GSJ was directly involved in the study in question and is a senior author on the paper referred here.

Reviewer 3 Report

This is a good review article. I have only minor suggestions.

(1) Line 252: authors can perhaps explain why cofactors did not clearly show up in the electron density map.

(2) Table 1 requires reformatting.

(3) Line 297: Works by Charles Weissmann could be cited. Eigen may have initiated the concept of "quasi-species", but it was Weissmann, who (correctly or not) applied it when discussing prion strains.

Author Response

Reviewer 3

We thank the reviewer for their helpful comments. We have amended the manuscript to incorporate their recommendations.

  • Line 252: authors can perhaps explain why cofactors did not clearly show up in the electron density map.

The reasons are either the cofactor(s) may not constitute part of mature prion or may not be rigid and homogenous in conformation and hence not definable using EM techniques. We have amended the manuscript to discuss this.

  • Table 1 requires reformatting.

We have reformatted this Table to improve its readability.

  • Line 297: Works by Charles Weissmann could be cited. Eigen may have initiated the concept of "quasi-species", but it was Weissmann, who (correctly or not) applied it when discussing prion strains.

We amended the manuscript to include this citation.

Round 2

Reviewer 2 Report

Overall this is a greatly improved manuscript. There are a few minor issues that the authors might want to check before publishing. Please see attached file.
